# SecNet: Semantic Eye Completion in Implicit Field

**Yida Wang**[*]                                    YIDA@FB.COM
**Yiru Shen**[†]                            SHENYIRUSTAR@GMAIL.COM
**David Joseph Tan**                          DJTAN@GOOGLE.COM
**Federico Tombari**                          TOMBARI@IN.TUM.DE
**Sachin Talathi**                              STALATHI@FB.COM

**Editor:** Editor's name

## Abstract

If we take a depth image of an eye, noise artifacts and holes significantly affect the depth values on the eye due to the specularity of the sclera. This paper aims at solving this problem through semantic shape completion. We propose an end-to-end approach to train a neural network, called *SecNet* (semantic eye completion network), that predicts a point cloud with an accurate eye-geometry coupled with the semantic labels of each point. These labels correspond to the essential eye-regions, i.e. pupil, iris and sclera. Particularly, our work performs implicit estimation of the query points with semantic labels where both the semantic and occupancy predictions are trained in an end-to-end way. To evaluate the approach, we then use the synthetic eye-scans rendered in UnityEyes simulator environment. Compared to the state of the art, the proposed method improves the accuracy for shape-completion for 3D eye-scan by 8.2%. In practice, we also demonstrate the application of our semantic eye completion for gaze estimation.

**Keywords:** Eye completion, Implicit Field, Semantic Completion

## 1. Introduction

Video-oculography (VOG) has gained popularity in recent years as a method for eye-tracking (van der Geest and Frens, 2002; Larrazabal et al., 2019; Nair et al., 2020). The main elements of VOG are egocentric cameras that capture images of eye, which then undergo image-processing techniques to extract the eye movement information. 3D VOG systems on the other hand also extract torsional eye-position using iris and pupil landmarks (Goni et al., 2004). As such, the accuracy of pupil tracking is central to the performance of VOG system, which can be significantly hampered by occlusions.

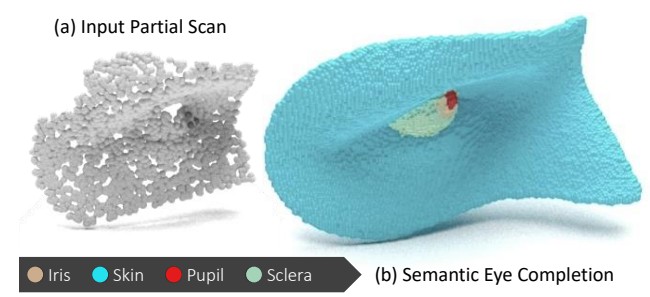

Figure 1: Given a partial scan of an eye in (a), our semantic completion in (b) reconstructs the fine-grained eye surface where each point is semantically labeled.

---

[*] Work was done during internship in Facebook Reality Labs.

[†] Work was done in Facebook Reality Labs.

Methods such as ellipse fitting (Fitzgibbon et al., 1999), RANSAC outlier removal (Jian and Chen, 2010) and moving average filtering (Satriya et al., 2016), and more advanced methods such as circular Hough transforms (Cherabit et al., 2012) for extreme pupil occlusions (Setiawan et al., 2018) have in particular been found useful to solve the pupil occlusion problems. However, in recent years, several algorithmic approaches that leverage 3D eye structures (Liu et al., 2021, 2020a) have been proposed for pupil tracking in the presence of occlusions.

Our work is focused on utilizing the 3D eye regions for pupil tracking. We leverage recent advances in 3D machine learning to reconstruct the precise 3D structure of the eye region to fill out the occluded regions. As shown in Fig. 1, shape completion is carried out on the partial scan of eye to fill out the occluded eye regions.

Several works in recent years have addressed the problem of 3D shape completion using learning based methods. These methods can be classified based on the data-format for 3D scans. The most popular data-formats include volumetric (Song et al., 2017; Dai et al., 2018), meshes (Groueix et al., 2018; Wei et al., 2021), point cloud (Chang et al., 2015; Dai et al., 2017a) and implicit representation (Park et al., 2019; Erler et al., 2020; Chibane et al., 2020). Among them, the implicit 3D reconstruction frameworks such as DeepSDF (Park et al., 2019), IF-Net (Chibane et al., 2020) and Points2Surf (Erler et al., 2020) provide high resolution 3D shape-completion by estimating the implicit values for random 3D query points. To estimate the implicit values, point-wise feature extractors such as PointNet features (Qi et al., 2017a) and PointNet++ features (Qi et al., 2017b) are commonly used.

To solve the efficiency issue caused by $k$-nearest neighbour search in the PointNet++ feature space, this paper adopts SoftPool (Wang et al., 2020b) feature as a local descriptor to construct an end-to-end model, called *SecNet*, to estimate implicit code for each given 3D query point. With the additional information of semantics used during training, our model is able to perform semantic completion in an implicit field of the eye region to precisely complete the eye surface geometries, coupled with semantics including the sclera, iris and pupil.

The training dataset for SecNet depends on the paired 2D partial scan and 3D ground truth. We create our dataset by synthesizing eye-scans using the UnityEyes (Wood et al., 2016) simulator. We then render the 3D eye-scans of 1,000 distinct people, each fixating on nine different gaze points as shown in Fig. 2.

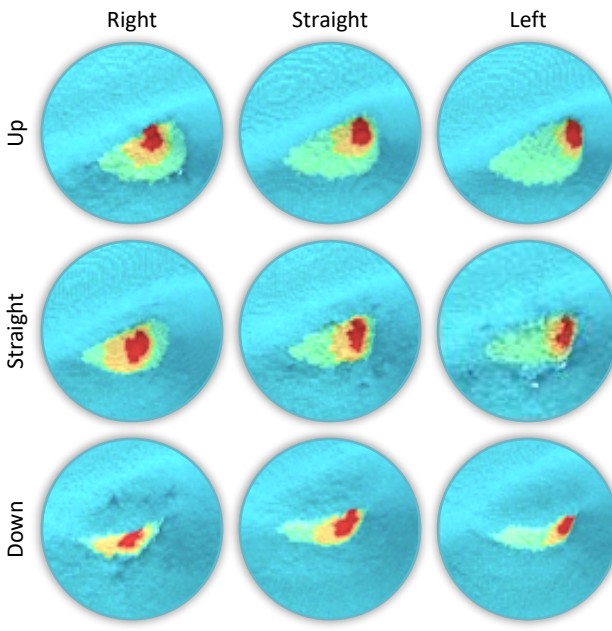

Figure 2: Nine gaze directions.

Empirically, our proposed the semantic implicit completion model is validated on this eye region dataset, achieving state-of-the-art performance at reconstructing semantic ge-

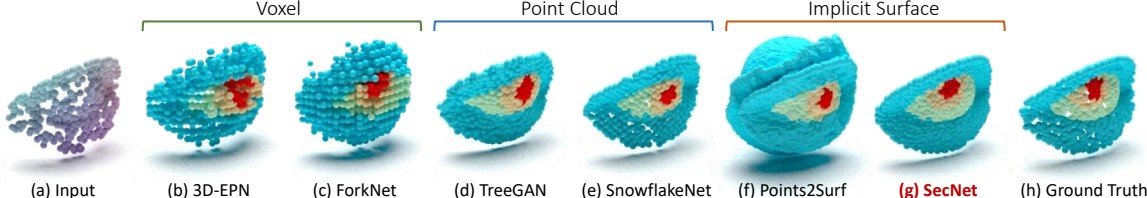

Figure 3: Given the input partial scan in (a) and the ground truth in (h), we compare different representations for semantic eye completion such as volumetirc data (b, c), point cloud (d, e) and implicit surface (f, g). Note that, except for (c) and (g) that directly infer the semantic completion, the approaches are segmented by 3D-GCN (Lin et al., 2020) to predict the semantic labels.

ometries. Moreover, we empirically demonstrate that the accurate reconstruction of the completed eye region is helpful for gaze estimations.

## 2. Related works

This section focuses on the more general related work on 3D completion and semantic completion. In addition, since we are proposing to use the semantic eye completion for gaze estimation, we also discussed the related works that relied on depth images to estimate the gaze direction.

### 2.1. 3D completion

There are three different ways of completing a shape from a partial scan as shown in Fig. 3: volumetric grid, point cloud and implicit surface. Early works using deep learning have relied on volumetic reconstruction because of its similarity to images, which allowed them to extend the convolution operation to 3D. For instance, 3D-EPN (Dai et al., 2017b) takes TSDF volumes (Werner et al., 2014) as input and builds an encoder-decoder structure using 3D convolution. SSCNet (Song et al., 2017) proposed to use flipped TSDF as input to perform semantic segmentation and completion at the same time. To solve the lack of 3D annotations, ForkNet (Wang et al., 2019) proposed to use the discriminator to synthetically generate new pairs of partial scan and its corresponding completed reconstruction. The main issue in such approaches is that storing 3D data in a dense volumetric grid (Song et al., 2017; Dai et al., 2018) consumes too much disk space and slows down inference speed for down-stream applications (Dai et al., 2018).

Point clouds were the less popular choice because of its unorganized structure. Notably, unlike volumetric data, we cannot easily apply the 3D convolution operations on them. To handle this issue, PointNet (Qi et al., 2017a) proposed a solution that uses max-pooling operations to make the feature permutation invariant so that the order of the points going through the architecture does not matter. Such feature was initially proposed for 3D object classification and segmentation, which was later used in point cloud completion in FoldingNet (Yang et al., 2018), PCN (Yuan et al., 2018) and AtlasNet (Groueix et al., 2018).

PointNet feature, however, lacks the ability to describe the local geometry in the point cloud. This motivated the extended version PointNet++ (Qi et al., 2017b) that uses $k$-nearest neighbor search to describe the local structure. SoftPoolNet (Wang et al., 2020b), on the other hand, is also motivated by the same objective but avoids running the time-consuming $k$-nearest neighbor search. Instead, this method proposes to use trainable parameters to sort the points through the feature dimension.

As we can observe in Fig. 3, completion with implicit surface generates smoother reconstruction with significantly less noise compared to volumetric and point clouds. Although their input is also based on point cloud features (Erler et al., 2020; Guerrero et al., 2018), implicit 3D reconstruction such as DeepSDF (Park et al., 2019), IF-Net (Chibane et al., 2020) and Points2Surf (Erler et al., 2020) creates a fine-grained 3D shape by estimating an object surface which distinguishes the inner and outer space. Such format not only produces smoother surface reconstruction, but also reveals more local structural details compared to traditional mesh reconstruction approaches such as screened poisson reconstruction (SPR) (Kazhdan and Hoppe, 2013).

### 2.2. Semantic completion

While several methods focus on completion alone (Dai et al., 2017b; Park et al., 2019; Dai et al., 2018; Yuan et al., 2018), there are other methods which simultaneously infer the semantic labels with the geometric completion (Song et al., 2017; Wang et al., 2019, 2018). For instance, SSCNet (Song et al., 2017) uses 3D dilated convolutions to build an encoder-decoder architecture to predict semantic and occupancy value for each voxel in a predefined 3D grid. ForkNet (Wang et al., 2019) proposes a decoder with three branches which are able to generate realistic newly paired partial scan and its semantic completion to train the entire network for semantic completion.

For methods that perform completion alone (Dai et al., 2017b; Wang et al., 2020b; Erler et al., 2020), one solution to gain semantic labels is to attach a segmentation framework, e.g. PointNet (Qi et al., 2017a) and PointCNN (Li et al., 2018), after the geometric completion. Recently, some features are proposed for point cloud segmentation to exploit local neighbourhood such as PointNet++ (Qi et al., 2017b) focusing on extracting features from local point groups. Also using the nearest neighbour search for feature extraction, KCNet (Shen et al., 2018) further aggregates the local features to investigate more complex relationships. KPConv (Thomas et al., 2019) and 3D-GCN (Lin et al., 2020) make the kernel of the point cloud convolution deformable to generate better matches with different local geometries for segmentation. Although implicit reconstruction is already well investigated, an implicit reconstruction with semantic estimation is not explored. In this paper, we are proposing one of the first few works on semantic implicit completion.

### 2.3. Gaze estimation

There are different ways to estimate the gaze, especially when using RGB images (Jianfeng and Shigang, 2014). However, in this work, we limit the scope to the utilization of 3D data which is less investigated.

To determine the direction of the user's gaze, one of the most important parameters is the position of the pupil, while the other is the eye's spherical center. Given the 3D structure,

EMGE (Zhou et al., 2016) proposes to estimate the entire spherical eyeball structure by estimating several points in pupil while RTGE (Sun et al., 2015) locates the pupil by fitting a circle to 2D scans which is back-projected to 3D for the gaze estimation. To the best of our knowledge, we are the first approach that uses semantic eye completion to perform gaze estimation.

With our solution, we encountered an issue in finding the appropriate dataset to train our models. The publicly available datasets do not provide the paried 3D partial scans and their completion. Most datasets focus on RGB images such as SynthesEyes (Wood et al., 2015) which synthesizes 2D eye images with realistic illumination. This then prompt us to build and publish a new dataset using the UnityEyes (Wood et al., 2016) simulator engine. Similar to ShapeNet (Chang et al., 2015) for objects and ScanNet (Dai et al., 2017a) for scenes, we then propose a method to build dataset based on eye meshes such as UnityEyes (Wood et al., 2016) specializing on semantic eye completion.

## 3. Methodology

The input to the framework is a partial scan captured by a depth camera pointing towards the eye. With $N_{\text{scan}}$ points, each with $(x, y, z)$ coordinate, we denote the partial scan as a point cloud $\mathcal{P}_{\text{scan}}$ which is represented as an $N_{\text{scan}} \times 3$ feature map. In practice, these scans are affected by noise and self-occlussions, e.g. from the eyelid and eyelashes. The objective then is to fix these issues and build a completed point cloud $\mathcal{P}_{\text{eye}}$.

Since semantic supervision is available for training, we also predict the semantic labels that includes the skin, sclera, iris and pupil. While some methods (Lin et al., 2020) utilize another inference model $\mathcal{S}$ such that the segmentation is predicted separately from the completion as $S_{\text{eye}} = \mathcal{S}(\mathcal{P}_{\text{eye}})$, we propose to use a single model to infer the semantic eye completion as $\mathcal{G}_{\mathcal{S}}(\cdot)$. This therefore estimates the geometry and the semantics at the same time.

### 3.1. Semantic implicit fields

Moreover, we take advantage of the particular problem at hand. Notably, the similarities of the eye structures across different individuals and different movements allow us to effectively use implicit reconstruction. This then solves the limitation from unstructured point cloud in terms of structural accuracy and the limitation from voxel grids in terms of reconstruction resolution; consequently, leading to a reduction of noise in the reconstruction with high resolution. This is validated in Fig. 3. In addition, we noticed that our method even produced a denser and smoother reconstruction than the ground truth.

Inspired by the works of IF-Net (Chibane et al., 2020) and Points2Surf (Erler et al., 2020), we also learn implicit values between a set of query points and the mesh surface. For each query point, these methods predict a value between $-1$ to $1$, where a query point on the surface is at the zero-crossing. The difference between traditional implicit surface learning and our model is that we propose to present the semantic labels in addition to the geometry, which we call *semantic implicit field* (SIF). Therefore, in our work, given an arbitrary query point $p_{\text{query}}$, we can simplify the framework to a classification task where the architecture predicts if the point is an *empty* space, or part of the *skin*, *sclera*, *iris* or *pupil*. We refer the classification result as the *semantic code* $c_{\text{query}}$ of the query point

such that $c_\text{query} = \mathcal{G}_\mathcal{S}(\mathcal{P}_\text{scan}, p_\text{query})$. This implies that the output eye reconstruction $\mathcal{P}_\text{eye}$ is presented by all the query points that are not empty. In practice, assuming that the partial scan is normalized to a unit cube, we sample the query points randomly around the partial scan within a Chamfer distance of 0.3.

### 3.2. SecNet architecture

The architecture for $\mathcal{G}_\mathcal{S}(\cdot)$ is summarized in Table 1, where we build an encoder-decoder structure. Here, the encoder processes the partial scan $\mathcal{P}_\text{scan}$ and produces the latent feature that describes the global structure. Having the latent feature and a query point $p_\text{query}$, the decoder runs the implicit estimation that finds the semantic code $c_\text{query}$ which classifies whether the point is empty or the specific part of the eye.

In particular, the encoder first randomly sub-samples the $\mathcal{P}_\text{scan}$ into $N_\text{in} = 2048$ points to have a constant tensor as input. These points are fed to a 3-layer MLP that generates an output dimension of 8. We then use the SoftPool (Wang et al., 2020b) operation with the number of regions $N_f = 8$ and the number of regional points $N_r = 32$, which produces an 8-region feature map with the shape of $[256, 8]$. This is processed by a regional convolution (Wang et al., 2020b) with a kernel size $D_\text{kernel} = [N_p, N_f]$ for all 8 regions, which covers $N_p = 8$ points with zero padding in each region. We then add a 2D convolution with kernel size $D_\text{kernel} = [N_r \times N_f, N_f]$, resulting in a 64-dimensional vector as the latent feature. Since the encoder is only dependent on the partial scan, the latent feature is constant across all the query points in the decoder.

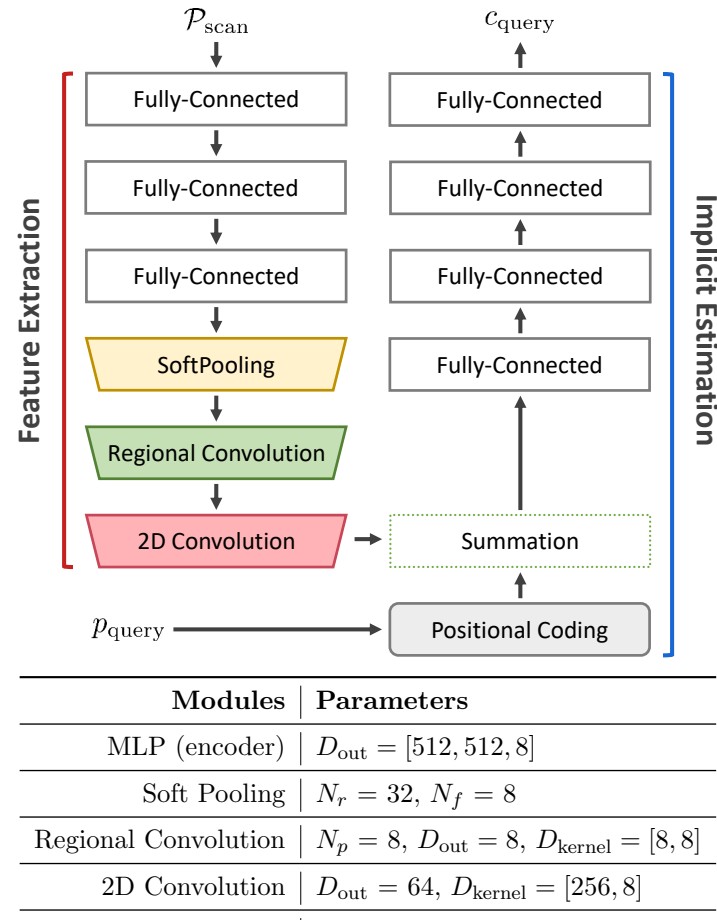

| Modules | Parameters |
|---|---|
| MLP (encoder) | $D_\text{out} = [512, 512, 8]$ |
| Soft Pooling | $N_r = 32$, $N_f = 8$ |
| Regional Convolution | $N_p = 8$, $D_\text{out} = 8$, $D_\text{kernel} = [8, 8]$ |
| 2D Convolution | $D_\text{out} = 64$, $D_\text{kernel} = [256, 8]$ |
| Positional Coding | $D_\text{out} = 64$ |
| MLP (decoder) | $D_\text{out} = [16, 32, 64, 5]$ |

Table 1: Architecture of SecNet with the corresponding hyperparameters for each module where $D_\text{out}$ represents the output dimensions.

Every query point in the decoder is converted to a positional code using SIREN (Sitzmann et al., 2020), having the same dimension as the latent feature. Thereafter, the sum of the positional coding and the latent feature serves as the input to the 4-layer MLP to estimate the final semantic code $c_{\text{query}}$ with a softmax activation.

We summarize the numerical values of our architecture in Table 1. This table shows the architecture on top and the corresponding parameters for each layer at the bottom.

To train the proposed encoder-decoder model, we impose the per-category binary cross entropy $\epsilon_c(\cdot, \cdot)$ such that

$$\mathcal{L}_{\text{semantic}} = \sum_c \epsilon_c(c_{\text{query}}, c_{\text{gt}}) = \sum_c \epsilon_c \left( \mathcal{G}_{\text{S}}(\mathcal{P}_{\text{scan}}, p_{\text{query}}), c_{\text{gt}} \right) \tag{1}$$

sums up the entropy for all categories. Given this loss function, we train the model $\mathcal{G}_{\mathcal{S}}$ with a batch size of 64. We employ the Adam optimizer (Kingma and Ba, 2015) with a learning rate of 0.0001 while the exponential decay rates $\beta_1$ and $\beta_2$ are set to 0.9 and 0.999, respectively.

### 3.3. Gaze estimation

As a by-product of our semantic eye completion, we estimate the gaze direction through the semantic points. Similar to RTGE (Sun et al., 2015), we solve this problem by estimating a 3D vector from the center of the eyeball to the center of the pupil. To find the centers, we use all the points on the sclera to fit a sphere that represents the eyeball; then, take the average point of all the points on the iris. The gaze direction is finally estimated as the vector that connects the center of the sphere and the average point. For the sphere, we use an energy optimization to estimate its center $p_{\text{center}} = (x_c, y_c, z_c)$ as well as its radius $r$. This minimizes the loss

$$\mathcal{L}_{\text{eyeball}} = \sum_i^{N_{\text{sclera}}} \left| \left\| p_{\text{sclera}}^i - p_{\text{center}} \right\|^2 - r^2 \right| , \tag{2}$$

summing up the absolute errors from all the $N_{\text{sclera}}$ points labelled as sclera $p_{\text{sclera}}^i$ in the semantic eye completion.

## 4. Dataset

We generate the dataset by rendering pairs of partial scans and their corresponding semantic completion using the UnityEyes (Wood et al., 2016) mesh models. To generalize for the gaze estimation, we rotate the eyeball towards nine gaze directions as shown in Fig. 2, including up-right, up, up-left, right, straight, left, down-right, down and down-left. With 1,000 identities from UnityEyes, we then have a total of 9,000 pairs in the dataset. For the experiments, we split our dataset with 800 identities for training and 200 for testing. One of the main advantages in having depth images or partial scans as input is the privacy preservation in training or during inference.

Fig. 4 shows some examples of the process that the model undergo when creating the dataset. Since the dataset is synthetically rendered, we impose the defects and self-occlusion by randomly dropping 61.2% points for pupil, 74.9% of iris, 29.7% of sclera and 9.0% of skin on every mesh models as shown in Fig. 4(b). In addition, the surface also incorporates jitter in order to mimic the sensor noise. The jitter is defined by a Gaussian distribution with zero mean and a 0.05 standard deviation for points on the sclera, iris and pupil. For this dataset, Fig. 4(c) illustrates the example input partial scans that we use for this evaluation. Noticeably, without visualizing the ground truth semantic labels in Fig. 4(c), iden-

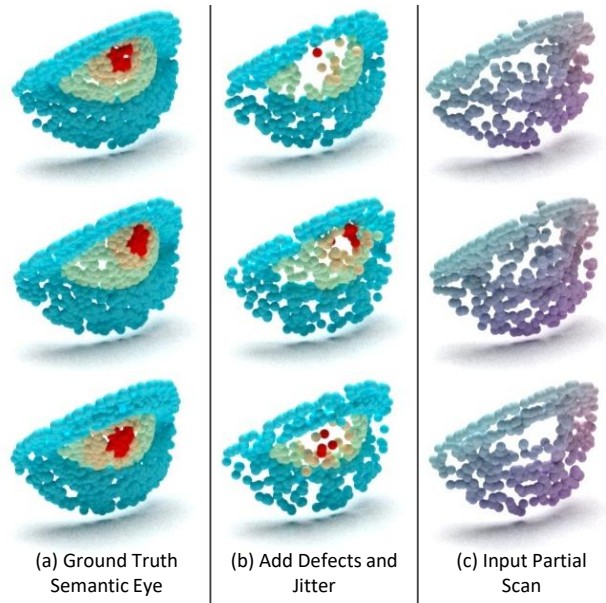

(a) Ground Truth Semantic Eye    (b) Add Defects and Jitter    (c) Input Partial Scan

Figure 4: Examples of our dataset.

tifying the regions on the eye or the gaze direction from the three images becomes very difficult.

## 5. Experiments

To highlight our contributions in this work, we conduct the following 3 experiments: synthetic eye augmentation, semantic eye completion and gaze estimation. We empirically demonstrate the advantages of our approaches using the dataset from Sec. 4.

### 5.1. Semantic eye completion

We evaluate the semantic completion on the eye region on the dataset captured from UnityEyes (Wood et al., 2016) models. The point clouds of the partial scans serves as the input for both training and testing. Their corresponding ground truth completed shape is presented in terms of a point cloud with semantic labels. Our evaluations are carried out across 4 categories including skin, sclera, iris and pupil. Note that, across all methods, the input partial scan is normalized into the same scale of point coordinates ranging between $-0.5$ to $0.5$.

This evaluation compares against the state-of-the-art completion methods that reconstruct using volumetric data, point cloud or implicit surface. While Fig. 3 shows the qualitative results of different approaches, Table 2 highlights the numerical comparison between them. This table shows that we achieve the state-of-the-art results, reaching an average L1-Chamfer distance of $6.15 \times 10^{-3}$ on all the categories. It also shows that we have the best results across all the four categories. It is noteworthy to mention that only ForkNet (Wang et al., 2019) and our approach perform semantic completion, while other methods can only infer the geometric completion. Due to this, for the other methods, we apply 3D-GCN (Lin et al., 2020) on the reconstruction to find the semantic labels.

| | Method | skin | sclera | iris | pupil | *avg.* |
|---|---|---|---|---|---|---|
| *Voxel* | 3D-EPN (Dai et al., 2017b) | 13.43 | 22.09 | 19.43 | 15.96 | 17.73 |
| | ForkNet (Wang et al., 2019) | 17.04 | 14.75 | 18.16 | 14.78 | 16.18 |
| *Point Cloud* | PointNet++ (Qi et al., 2017b) | 9.72 | 10.24 | 12.73 | 11.85 | 11.13 |
| | FoldingNet (Yang et al., 2018) | 9.35 | 10.23 | 12.29 | 11.68 | 10.89 |
| | TopNet (Tchapmi et al., 2019) | 8.82 | 10.22 | 11.82 | 11.02 | 10.48 |
| | AtlasNet (Groueix et al., 2018) | 8.15 | 9.70 | 11.18 | 10.64 | 9.92 |
| | PCN (Yuan et al., 2018) | 7.48 | 9.69 | 11.12 | 10.29 | 9.65 |
| | MSN (Liu et al., 2020b) | 7.01 | 9.10 | 10.63 | 9.77 | 9.13 |
| | SoftPoolNet (Wang et al., 2020b) | 6.54 | 8.78 | 9.79 | 9.46 | 8.65 |
| | GRNet (Xie et al., 2020) | 6.22 | 8.70 | 9.68 | 9.14 | 8.44 |
| | PMP-Net (Wen et al., 2020) | 5.74 | 8.26 | 8.98 | 8.83 | 7.96 |
| | CRN (Wang et al., 2020a) | 5.57 | 8.23 | 8.98 | 8.81 | 7.90 |
| | SnowflakeNet (Xiang et al., 2021) | 4.93 | 7.48 | 8.76 | 8.68 | 7.46 |
| *Implicit* | IF-Net (Chibane et al., 2020) | 5.43 | 7.98 | 7.95 | 7.09 | 7.12 |
| | Points2Surf (Erler et al., 2020) | 4.93 | 7.45 | 7.48 | 6.91 | 6.70 |
| | **SecNet** | **4.21** | **6.99** | **7.17** | **6.25** | **6.15** |

Table 2: Evaluation of the semantic eye completion. We measure the Chamfer distance for each category; and, compute the average across all categories.

If we investigate closely on the comparison against the volumetric methods, their error are significantly higher, e.g. ForkNet (Wang et al., 2019) has an average Chamfer distance of $16.18 \times 10^{-3}$, since they only use a grid with an output dimension of $[64, 64, 64]$. To evaluate in the same metric, we convert the grid into a point cloud before evaluating the Chamfer distance. From Fig. 3, we can observe that they have an obvious disadvantage due to their low resolution.

As for point cloud approaches, we compare our semantic eye completion results with some recently proposed methods such as FoldingNet (Yang et al., 2018), PCN (Yuan et al., 2018), MSN (Liu et al., 2020b), GRNet (Xie et al., 2020) and SoftPoolNet (Wang et al., 2020b). All these approaches reconstructs a point cloud which is further re-sampled to 4,096 points for evaluation. Compared to SnowflakeNet (Xiang et al., 2021) which is the state-of-the-art point cloud completion approach, our method achieves decreased the Chamfer distance by $1.31 \times 10^{-3}$.

Lastly, when we compare against other implicit method, Points2Surf (Erler et al., 2020) also performs well with a Chamfer distance of $6.70 \times 10^{-3}$. However, our proposed architecture performs the best among all listed approaches in Table 2 with an error of $6.15 \times 10^{-3}$.

## 5.2. Gaze estimation

Since we can convert the semantic eye completion to gaze direction through Sec. 3.3, this section focuses on the evaluation of the gaze direction. In addition to the gaze from semantic completion, we also include the related work that designed for gaze estimation such as

| | Method | Accuracy | Cosine Distance | Model Size (MB) | Time (seconds) |
|---|---|---|---|---|---|
| Direct Gaze Estimation | EMGE (Zhou et al., 2016) | 42.1% | 0.637 | – | 0.14 |
| | RTGE (Sun et al., 2015) | 56.9% | 0.691 | – | 0.08 |
| | 3D-GCN (Lin et al., 2020) | 61.8% | 0.745 | 6.6 | 0.82 |
| Voxel | 3D-EPN (Dai et al., 2017b) | 81.4% | 0.802 | 420.0 | 0.82 |
| | ForkNet (Wang et al., 2019) | 83.8% | 0.809 | 362.0 | 1.12 |
| Point Cloud | PointNet++ (Qi et al., 2017b) | 82.9% | 0.781 | 29.7 | 2.33 |
| | FoldingNet (Yang et al., 2018) | 84.6% | 0.807 | 19.2 | 0.05 |
| | TopNet (Tchapmi et al., 2019) | 85.1% | 0.822 | 79.9 | 0.61 |
| | AtlasNet (Groueix et al., 2018) | 85.2% | 0.821 | 2.0 | 0.32 |
| | PCN (Yuan et al., 2018) | 87.3% | 0.823 | 54.8 | 0.11 |
| | MSN (Liu et al., 2020b) | 88.0% | 0.830 | 12.0 | 0.21 |
| | SoftPoolNet (Wang et al., 2020b) | 89.2% | 0.842 | 37.2 | 0.04 |
| | GRNet (Xie et al., 2020) | 91.6% | 0.857 | 293.0 | 0.88 |
| | PointCNN (Li et al., 2018) | 87.6% | 0.826 | 497.0 | 1.20 |
| | PMP-Net (Wen et al., 2020) | 90.6% | 0.850 | 22.0 | 4.21 |
| | CRN (Wang et al., 2020a) | 93.1% | 0.884 | 61.5 | 2.73 |
| Implicit | IF-Net (Chibane et al., 2020) | 93.6% | 0.909 | 29.4 | 9.27 |
| | Points2Surf (Erler et al., 2020) | 94.3% | 0.921 | 24.0 | 12.64 |
| | **SecNet** | **97.6%** | **0.971** | 9.7 | 0.19 |

*(Left spanning labels: "Gaze from Semantic Eye Completion")*

Table 3: Evaluation of the gaze direction classification and estimation with the corresponding model size and inference time. The table is divided into two regions. The methods on top directly use the depth image to find the gaze; while, the methods at the bottom estimates the gaze based on the semantic eye completion. Note that (Zhou et al., 2016; Sun et al., 2015) does not depend on a parameterized inference model.

EMGE (Zhou et al., 2016) and RTGE (Sun et al., 2015). These methods directly locate the pupil and estimate the center of eyeball from 2D partial scan without eye completion. We also include 3D-GCN (Lin et al., 2020) which segments the input partial scan into parts prior to the gaze estimation.

We first consider this as a classification problem where we match the estimated gaze based on the nine directions. Comparing with other methods in Table 3, our approach reached a classification accuracy of 97.6% which is significantly higher than any other approach.

Instead of relying only on discrete values, we also considered the cosine distance to evaluate the estimated gaze from the ground truth, which is the dot product of the two vectors. Here, our approach also produces the best performance of 0.971.

### 5.3. Efficiency

We also evaluate the processing time and the corresponding memory footprint of each model, which is summarized in Table 3. This table illustrates that our inference time at 0.19 seconds is much faster than the other implicit reconstruction methods such as DeepSDF (Park et al., 2019) at 9.72 seconds and Points2Surf (Erler et al., 2020) at 12.64 seconds. This is because the other methods require zero-crossing in reconstruction while our method does not. In addition, our point-wise implicit estimation is conditioned on a SoftPool feature from the encoder, which is processed once for each partial scan. Points2Surf, on the other hand, decodes the implicit values using QSTN (Guerrero et al., 2018) which depends on analyzing the local point cloud patches. This implies that it needs to be executed repetitively for each query point. As a consequence, we can reduce the time by focusing on areas surrounding the partial scan to decrease the number of query points to process. Overall, we do not attain the lowest memory footprint or the lowest inference time. However, we argue that our approach has a good trade-off between the two parameters.

### 5.4. Ablation study

We perform an ablation study to understand the effect of changes in the hyperparameters. Table 4 summarizes this study. We noticed that the MLP in the encoder does not significantly change the completion performance as long as the output dimension of the first layer of the MLP is larger than 256. Having values above 512 only improves the average Chamfer distance from $6.15 \times 10^{-3}$ to $6.09 \times 10^{-3}$.

Since our latent feature is extracted by SoftPool (Wang et al., 2020b) operators, we validate the changes in performance by adapting different input feature dimension $N_f$ and number of points $N_r$ chosen from each of the sequential feature map. Our experimental results show that the performance ranges from $6.10 \times 10^{-3}$ to $6.84 \times 10^{-3}$. This is validated by (Wang et al., 2020b) where we can reach a good performance as long as the feature dimension $N_f$ is larger than 4. For the MLP in decoder, we found that performance saturates when $D_{\text{out}} = [16, 32, 64, 5]$.

## 6. Conclusion

In this paper, we propose to complete the eye region through *semantic implicit field*. Using our semantic eye completion, we also introduce a more practical use-case, i.e. gaze estimation. We achieve the state-of-the-art performance for both semantic eye completion and gaze estimation. Since we propose a new problem in semantic completion and a new type of solution for gaze estimation, we propose a simple way to build the dataset for semantic completion eyes based on UnityEyes (Wood et al., 2016) meshes to train and evaluate the models.

| Modules | Parameters | Chamfer Distance |
|---|---|---|
| positional coding | Gaussian (Tancik et al., 2020) | 6.42 |
| | SIREN (Sitzmann et al., 2020) | **6.15** |
| | sinusoidal (Vaswani et al., 2017) | 6.31 |
| MLP (encoder) | $D_{\text{out}} = [512, 256, 8]$ | 6.99 |
| | $D_{\text{out}} = [256, 512, 8]$ | 7.33 |
| | $D_{\text{out}} = [512, 512, 8]$ | **6.15** |
| | $D_{\text{out}} = [1024, 512, 8]$ | 6.12 |
| | $D_{\text{out}} = [512, 1024, 8]$ | 6.09 |
| Softpool (Wang et al., 2020b) | $N_r = 16, N_f = 8$ | 6.84 |
| | $N_r = 32, N_f = 4$ | 8.07 |
| | $N_r = 32, N_f = 8$ | **6.15** |
| | $N_r = 32, N_f = 16$ | 6.10 |
| | $N_r = 64, N_f = 8$ | 6.14 |
| MLP (decoder) | $D_{\text{out}} = [16, 16, 64, 5]$ | 7.04 |
| | $D_{\text{out}} = [16, 32, 32, 5]$ | 6.60 |
| | $D_{\text{out}} = [16, 32, 64, 5]$ | **6.15** |
| | $D_{\text{out}} = [16, 64, 64, 5]$ | 6.13 |
| | $D_{\text{out}} = [16, 32, 128, 5]$ | 6.11 |

Table 4: Ablation study on network hyperparameters. The results in bold indicate the chosen parameters in architectural design which balance the accuracy and model size.

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
