# OpenReview forum: "SecNet: Semantic Eye Completion in Implicit Field"
_NeurIPS.cc/2022/Workshop/GMML — Gaze Meets ML 2022 Oral_

### Official Review · Reviewer_ptUx · 2022-10-17
**A novel take on Gaze estimation**

**Rating:** 9
**Confidence:** 3

**Review:**

The authors present a novel approach to reconstructing the complete the eye region from a noisy partial eye scan. They describe the short comings in existing RGB based approaches and point out that 3D semantic surface modeling can lead to a more practical use-case, i.e. gaze estimation. The authors describe their methods and evaluation in sufficient detail and show that they achieve excellent performance for both tasks. They also propose a simple way to build a dataset for semantic completion eyes based on UnityEyes meshes.

The manuscript is relatively easy to read and makes arguments relatively well. They present evaluation results that are compelling in terms of performance and time, and in case of gaze estimation - accuracy.

Overall, I found the manuscript to be very good.

---

### Official Review · Reviewer_djjh · 2022-10-17
**Interesting idea for using a intermediate model of the eye to track, tested with synthetic data set.**

**Rating:** 5
**Confidence:** 3

**Review:**

The idea of creating an intermediate model of the eye for tracking is an interesting one. The weakest part of the paper seems to be the evaluation and lack of real world testing. With such validation the paper would be much more compelling and much stronger. Some specific questions:

* You say you are doing 8.2% above SotA. If you could convert this into a real world number (e.g., X degree tighter track).
* It seems like you are testing 9 possible gaze extrema - but this doesn't give a really good idea how well tracking would occur with a real world application.
* Is your statement about "too much disk space and slow interface speeds" a factor of the hardware you are running on? Is it something Moore's law will address, or is there something fundamental to the problem.
* You are modeling occlusion as a random stipple noise pattern. This is not really accurate is it? Occlusion is typically a group of pixels. The random noise is actually the best possible case for reconstruction. Did you test with larger drop-out regions?
* In your efficiency arguments; frame to frame tracking is typically a perturbation problem. How do you perform when you know the last frame?

---

### Meta-Review · Area_Chair_wA13 · 2022-10-20

**Recommendation:** Accept (Oral)
**Confidence:** 4

**Metareview:**

This work proposes a neural architecture point-cloud approach for reconstructing the eye geometry based on noisy partial observations. Reviewers have found the problem addressed interesting and the experimental results relatively compelling. Some of the reviews raised several questions that can be adequately addressed in the camera-ready version. Overall, the paper has received positive feedback that suggests acceptance.

---

### Decision · Program_Chairs · 2022-10-20

Accept (Oral)